



# Intense precipitation events in the Central range of the Iberian Peninsula

Manuel Mora García[1], Jesús Riesco Martín[2], José Miguel Sánchez Llorente[3], Luis Rivas Soriano[3], Fernando de Pablo Dávila[3]

[1]Spanish Meteorological Agency (AEMET), Oviedo, Spain

[2]Spanish Meteorological Agency (AEMET), Malaga, Spain

[3]Department of Fundamental Physics, University of Salamanca, Salamanca, Spain

*Correspondence to*: Fernando de Pablo Dávila (fpd123@usal.es)

**Abstract.** Intense orographic precipitation associated with the Central range was analysed using data from 19 episodes, with the highest average values for the study area, of precipitation accumulated within 24 h, occurring between

years 1958 to 2010. All events were associated with a south-westerly tropospheric flow, a low level jet, and high moisture flux at low levels. The observed moisture flux was higher than 100 m g (s kg)$^{-1}$ and the dry and wet Froude numbers were greater than 1. The selected area to study this synoptic situation was Gredos, broad and high range, which is located in the eastern part of the Central mountain range and generates leeward "orographic shadow". The effect of the Central range on the spatial distribution of precipitation on the Iberian Peninsula plateau results in a sharp increase in precipitation in the

south of the Central mountain range, followed by a decrease to the north of this range.

## 1. Introduction

The factors affecting the development of precipitation are complex and the forecasting of precipitation is therefore difficult and also increases with the use of larger spatial and temporal resolutions (Llasat and Siccardi, 2010).

Heavy precipitation is generally associated with a high moisture content, vertical movement, and static instability (Reale and Lionello, 2013; Chen et al., 2013). There are many studies addressing cases of heavy precipitation. Thus, for example, Schwartz *et al.,* (1990) analysed the evolution of the convective environment during episodes of heavy precipitation, and Fernández-Montes *et al.,* (2014) or Garavaglia et al., (2014) studied the synoptic patterns typically associated with heavy precipitation.

Some authors have attempted to elucidate the key factors involved in the production of heavy precipitation or intense orographic precipitation. These factors include precipitable water (Bližňák *et al.,* 2014), static instability (Funk, 1991), moisture at low levels (Massari et al., 2014), waves upstream from the area of heavy precipitation (Yu et al., 2015), dynamically forced vertical movements associated with wind maxima in the high troposphere (Ma and Bosart, 1990). Other articles have analysed moisture flux convergence at low levels (Banacos and Schultz, 2005), orographic lifting of

conditionally unstable air masses (Lin et al., 2001), and the fact that the maximum precipitation rate mainly depends on the ratio of mountain height to the level of free convection, the ridge/aspect ratio, and a parameter that measures the ratio of advective to convective time scales (Miglietta and Rotunno, 2009).





Research efforts have also focused on the study of heavy precipitation episodes in different areas. For example, Chiao et al., (2004) or Foresti and Pozdnoukhov (2012) discuss the heavy orographic precipitation in the Alps; Lang and Barros (2004) studied winter storms in the Central Himalayas, and Prat and Barros (2010) studied the spatial gradients and vertical structure of orographic precipitation in the southern Appalachians. Yu and Cheng (2008) analysed heavy

orographic precipitation associated with typhoon Xangsane. Furthermore, other such examples can be found in the works of Grumm et al., (2002) and Schumacher and Johnson (2006) studying the USA; Chen et al., (2007) studying Taiwan; Teixeira and Prakki (2007) studying Brazil; Federico et al., (2008) and Buzzi et al., (2014) studying Italy; Stefanescu *et al.,* (2014) for Romania or Dasari and Salgado (2015) for Madeira Island (Portugal).

In the case of Spain, heavy precipitation episodes are characteristics of the weather, especially on and around the

Mediterranean coast and during late summer and autumn (Riesco *et al.,* 2013). Jansa *et al.,* (2001) reported that approximately 90% of heavy precipitation episodes in the Internal Basins of Catalonia between 1996 and 2002 were associated with warm and moist flows at low levels generated by extratropical cyclones. Rigo and Llasat (2004) show that during the period 1996-2000, forty-three heavy rainfall events have been detected in Catalonia (Northeastern of Spain) and the most of these events caused floods and serious damage. Ramis *et al.,* (2009) studied heavy precipitation in the western

Mediterranean and found that deep convection was the main cause. Several studies have classified the meteorological patterns associated with heavy precipitation. Thus, Romero *et al.,* (1999) found the average geopotential field at the 925 hPa and 500 hPa pressure levels to be associated with heavy precipitation, and Martin-Vide (2002) classified the days with precipitation higher than 200 mm on the Spanish Mediterranean coast. Riesco *et al.,* (2013) reported that severe precipitation episodes in the southern Iberian Peninsula may be classified in three types according to moisture flux at the

850 hPa pressure level and the lifted index. All of these studies have analysed precipitation occurring in the same area and hence we believe it is necessary to explore other areas of the Iberian Peninsula where heavy precipitation also falls. Fernandez-Montes *et al.,* (2014) reported that a great amount of precipitation in the Iberian Peninsula concentrates in relatively few days, primarily conditioned by the atmospheric circulation and the moisture content, and they investigated the relationship between synoptic circulation types (CTs) and the frequency of precipitation extremes (> 90th percentile) in

spring and autumn at 44 different stations. Merino et al., (2016) conducted an analysis of precipitation extreme events (PEEs) in Spain between 1960 and 2011. Thresholds for determining event severity were defined using 99th percentiles, regions of extreme weather risk were identified and then trends of extreme precipitation index were analysed using the Mann–Kendall test.

The use of information obtained from statistical or case studies of precipitation episodes obtained from post-event

investigation is needed to improve the forecasting of precipitation and is the basis of effective operative warning for flooding due to extreme heavy precipitation, such as that based on the European precipitation index (Alfieri and Thielen, 2015). In the present study we analysed episodes that featured more intense orographic precipitation and also had high





instability index values during years 1958 to 2010, which affected the Central range of Iberian Peninsula but specifically those that occurred in the southern part of the area. This work is supplemented by including a study of meteorological indices that can help in the characterization of intense orographic precipitation. Finally, we analyse in detail two examples of episodes that demonstrate the direct application of the indices described above and that outline the difficulties involved

in the study of orographic precipitation.

## 2. Theoretical concepts

The effect of orography on air flow is a relevant issue in studies carried out at mesoscale, and air flow can be classified as: blocked flow, or flow that cannot pass over mountains; partially blocked flow, and flow that does pass over

mountains. To determine this, the Froude number can be used in the analysis of the characterization of the flow. The Froude number represents the quotient between the kinetic energy of the flow with respect to its potential energy, and its value is lower than 1 in the first case (subcritical flow), around 1 in the second case (critical flow), and much greater than 1 in the third case (supercritical flow). It is directly proportional to wind speed and inversely proportional to the height of the obstacle. In addition, moisture stability is taken into account to calculate the wet Froude number ($F_w$). This index is

important because the presence of moisture in the flow and the latent heat released through condensation reduce static stability and it may happen that although the dry Froude number is less than 1, and hence a blocked flow would be expected, the wet Froude number is greater than 1, and in that case, the flow is capable of passing over the orographic barrier (Durram and Klemp, 1982).

Several theoretical studies have been conducted to address the influence of the wet Froude number, the

convective available potential energy and mountain shape on conditionally stable flows crossing mountain barriers. For example, Smith (1979) and Colle (2004) investigated the influence of parameters such as wind, stability, moisture, mountain geometry, isozero height, etc. on the precipitation generated by moist flows over mountain barriers. Chu and Lin (2000) performed simulations of conditionally unstable flow over a two-dimensional mountain barrier and established three types of flow regimes as a function of $F_w$ :

• Type I ($F_w$ small): a convective system of up-stream propagation.

     • Type II ($F_w$ moderate): a long-lasting convective flow over the mountain.

     • Type III ($F_w$ large): a convective system over the mountain that propagates similarly to the flow.

In principle, the greatest windward precipitation would occur with moderate or small wet Froude number. In a

three-dimensional simulation, Chen and Lin (2001) reported that the presence of a jet at low levels exacerbates these windward precipitations. Chen and Lin (2005) classified flow as a function of $F_w$ and CAPE (Convective Available



Potential Energy), in agreement with the three types of flow described above, but added a fourth type (Type IV) that represents an orographic stratiform system, which probably propagates downstream.

The shape of the mountain barrier is defined by its height (h) divided by its half-width, which determines the distribution of the orographic precipitation (Smith, 1979). In the case of wide mountains, the maximum precipitation tends

to be windward (Rauber, 1992) to increase to higher areas, but leeward for narrow mountains (Sinclair *et al.,* 1997). For intense mean winds and a stable flow, in mountains with the same shape, the precipitation is usually intense on the highest and widest parts (Colle, 2004). Chen *et al.,* (2008) studied the influence of $F_w$ and mountain shape on flow. In their simulation with a numerical model, they varied wind speed at low levels and the half-width of the mountain, keeping mountain height (2000 m) and the CAPE (3000 J kg$^{-1}$) fixed.  They concluded that strong precipitations occur (200 mm in

10 h) and that these could generate flash floods or overflows with Type II flows during a significantly long period of time, or even with Type IV flows. These two situations have stratiform cloudiness rather than a convective type, although they are still capable of generating large amounts of precipitation comparable to convective cloudiness. The abundant moisture supplied could be the result of strong winds at low levels (for example, a jet stream).  They also reported that when $F_w$ is high and is kept fixed, the flow passes to a regimen of propagation in favour of the flow as the slope increases. When the

CAPE declines, it is assumed that the flow moves at a higher regimen. Later we will describe our findings with respect to these approaches.

Lin *et al.,* (2001) considered that if horizontal wind is calculated close to the mountain during the event, the non-linear contribution referring to the orographic vertical speed of the approaching synoptic system (for example, a short wave) could be partially contained in the horizontal wind. With some additional simplifications, they defined the index $U(\delta h/\delta x)q$

( with U the meridional component of wind velocity, $\delta h/\delta x$ the slope and q the mixing ratio) and concluded that with strong precipitation events in the Alps this value is greater than 4.7.

**3. Study area and data**

The study area is focused on the eastern part of the Central range, comprising the Gredos range (SGRED, Figure 1). This mountain range separate valleys of the Duero basin from those of the Tajo basin and have a maximum altitude of

2592 m and an average elevation of 1200 m.a.s.l.  A highly irregular relief characterizes this zone and the barrier is not uniform in height, width or orientation, such that from the point of view of index calculation many simplifications must be made. The eastern part of the Central range comprises the Gredos range, which forms the massif of the Central range and is very wide. In addition, most of the Gredos range, where the south slopes are much more abrupt than the leeward side, configure the Central range separating the southern plateau, with a lower altitude (about 400 m.a.s.l) from the northern

plateau (800 m.a.s.l).

From the climatological data network, we selected the 19 episodes with the heaviest precipitation (> 100 mm) in the western part of the Central range during the years 1958 to 2010 (Table 1). Additionally, it must be noted that episode



19 was included as a case study despite having a maximum precipitation slightly less than 100 mm, because is used us example of the importance of the time duration of the rain episode. The data regarding precipitation were obtained from the climatological database of the Spanish Meteorological Service [Agencia Estatal de Meteorologia (AEMET)]. They are world-class observatories equipped with staff, automatic stations and meteorological collaborators and they are subjected

to various quality controls. Data for the other meteorological fields were obtained from the ERA-40 database of the European Centre for Middle Range Weather Forecasting (ECMWF) for the 1958-2002 period  (Uppala *et al.,* 2005) and directly from the ECMWF model for the 2003-2010 interval. (http:// apps.ecmwf.int/datasets/data/era40).

The following parameters were calculated for each precipitation episode: mean and maximum precipitation; meridional moisture flux at the 850 hPa pressure level (obtained by multiplying the 850 hPa meridional component of

wind velocity and the specific humidity, $Vq$); the dry and moist Froude numbers (calculated as $F = V(Nh)^{-1}$, with $V$ the meridional wind, $N$ the dry or moist Brunt-Vaisala frequency, and $h$ the obstacle height); Convective Available Potential Energy (CAPE) and the Total Totals and K instability indices. It is worth mentioning here that we used the meridional wind component instead of the perpendicular component. The Central range is clearly west-east oriented, so we believe that the wind meridional component is an appropriate estimate of the flow in the mountains and therefore our calculated

values can be considered estimates of the Froude number. In the present study, the above-mentioned parameters were calculated (Kriging with ArcGIS© software) at 40 ºN, 5 ºW (belonging to the Gredos range area, Figure 1b), where southerly winds must surpass a height difference of ~ 1,500 m, and the Central range reaches its maximum altitude (2592 m.a.s.l.), and the maximum slope ($\Delta h/\Delta x$) is ~ 0.09 with a mean value in the southern hills of 0.05.

**4. Results and discussion**

The values of the meteorological parameters considered in this study for each precipitation event are shown in Table 1. Figure 2 shows the composite fields at 12 UTC of sea level pressure, wind and specific humidity at the 850 hPa pressure level and geopotential at the 500 hPa pressure level. These maps were built using data from the 19 episodes of severe rainfall indicated in Table 1. Figure 2a shows a low-pressure system at the sea level located on the north-western Iberian Peninsula with south-westerly winds over the Central range (Figure 2b). This is a moist flow and thus, increased

the humidity over the study area, especially on the south slopes, as shown in Figure 2c. The figure also shows a tongue of moist air at low levels over the south-western Iberian Peninsula, which reached values greater than 6 g kg$^{-1}$ over the study area. Moreover, a high wind speed was observed at the 850 hPa pressure level (Figure 2b), where a low-level jet stream, with a wind speed greater than 20 m s$^{-1}$, was apparent. A north-south oriented mid-level trough located to the west of the Iberian Peninsula (see Figure 2d) and moderate values of the instability indices (Table 1) were other common

characteristics associated with the heavy precipitation events studied in this work. The average values in the mean and maximum meridional moisture fluxes were 110 and 147 m g(s kg)$^{-1}$ respectively, and in most cases the individual values were higher than 100. As mentioned, Lin *et al.,* (2001) defined the $U(\Delta h/\Delta x)q$ index and reported that heavy orographic



precipitation in the Alps was associated with values higher than 4.7 at low levels. When this index was calculated for the Central range using the data in Table 1, the mean value was 5.5. Thus, this index also indicates the risk of heavy orographic precipitation for the Central range. Regarding the dry and wet Froude numbers, these ranged between 0.7 and 1.9, the maximum averaged values being 1.2 and 1.3 respectively, corresponding to moderate values of this index. The

shape of the mountain can be approached by using a slope of 1.6 km/35 km of the windward mountain barrier, (although at a resolution of 5 km there are maximum slope values of 0.09). With these values and in nearly all cases a Type III flow would be obtained which implies the presence of a convective system. However, it should be taken into account that with low CAPE and moderate wind intensity the flow type would be type IV instead of type III (Chen and Lin, 2005) giving rise to stratiform precipitation. In any case, both convective and stratiform precipitation would give large amounts of total

precipitation (including possible flooding), if the flow persists for many hours.

Table 2 shows the precipitation data of the Gredos range (SGRED) (Figure 1b). Colle, (2004) reported that with moderate winds (~ 20 m s$^{-1}$) and a freezing level over 750 hPa, the drop in the freezing level increases as mountain barrier height and width increase. Moreover, mountain waves when the wind is strong, static stability low and the mountain barrier narrow, may cause intense vertical movement. The orographic and precipitation profile across the SGRED is

depicted in Figure 3. This profile was calculated following the line AB indicated in Figure 1b. Figure 3 shows that precipitation over the SGRED increases sharply windward and that maximum values are reached on the upper third of the windward slope. Precipitation decreases sharply on the leeward slope, indicating an "orographic shadow" associated with the SGRED. Maximum precipitation is now located around the summit.

*a.- Case study:23-25 November 2006*

During this time frame (episode nº16), the precipitation observed at several stations along the Central range was more than 300 mm. Figure 4a shows the infrared channel image from the Meteosat satellite and Figure 4b shows the surface pressure pattern for November 24 at 00 UTC. A low-pressure system was affecting the North Atlantic area, with a minimum sea level pressure of ~ 970 hPa located west of Ireland. A south-westerly moist air flow and several frontal

systems were affecting the western Iberian Peninsula, moving from west to east. Figure 4c shows a deep north-south-oriented trough at the 300 hPa pressure level located west of the Iberian Peninsula, with a south-westerly low level jet of 60 m/s over the Central Iberian Peninsula. A south-westerly low-level jet stream of 30 m/s was also observed crossing the Central range at the 850 hPa pressure level on 25 November at 00 UTC (Figure 4d), which may have contributed to increase the precipitation (Chen and Lin, 2001). This meteorological situation generated a high-moisture flux (see Table 1)

and had a value of 7.2 for the $U(\Delta h/\Delta x)q$ index, clearly higher than the threshold value reported by Lin *et al.,* (2001) for the Alps. The cloud-to-ground lightning flashes observed on 24 November are shown in Figure 5a. The discharges were mainly located windward of the Central range indicating that flow was unstable. This is confirmed by the Madrid sounding



(40 ºN, 3 ºW) at 12 UTC (Figure 5b) and the values of the instability indices (Table 1). The Froude number is large, both for dry and moist cases, indicating a Type III flow regime. The remote-controlled observation station located in the western Central range and indicated by a circle in Figure 1b recorded 157.7 mm from 00 to 24 UTC of 24 November and 16.8 mm for 25 November. The temporal distribution on 24 November is shown in Figure 6a, and indicates that large

amounts of precipitation occurred regularly during that day. The pattern of the spatial distribution of average precipitation (1971-2000) (Figure 6b) shows the relevance of orographic precipitation in this area of the Iberian Peninsula.

Figure 7 shows the orographic profile and the precipitation along a south-north-oriented line crossing the SGRED. Precipitation decreased leeward of the broadest part of the Central range, and returned in large amounts windward of the Cantabrian range, located near the northern Spanish coastline (see Figure 1a). This marked "orographic

shadow" is consistent with the shape (resembling a C) of the spatial pattern of precipitation shown in Figure 6b. When the line was displaced to the west, the "orographic shadow" was not seen.

*b.- Case study: 27 February 2010*

The time duration of the rain episode is an important factor. Large amounts of precipitation are not usually associated with

episodes of short duration and this was the case of episode nº 19. On 27 February 2010, the extratropical cyclone "Xinthya" crossed the northwestern part of the Iberian Peninsula. This was an event of explosive cyclogenesis, with hurricane-strength winds. It caused heavy rain and local floods in the west of Spain (Hickey, 2011). Figure 8a shows the surface pressure pattern at 12 UTC. The fast-moving deep low of 976 hPa was located close to the northern coastline of Portugal, resulting in strong southerly winds impinging on the Central range. The same flow pattern was seen at medium

and high levels (not shown). Doppler radar (Figure 8b) shows a south-westerly low-level jet stream over the area studied.

Table 3 shows the sudden increase in the instability indices, Froude number and meridional moisture flux during the morning and central hours of 27 February, together with the low values before and after the event. This was due to the short duration of the flow perpendicular to the mountain range (Figure 9a). Therefore, the expected amount of rain cannot have been large. This is confirmed by the data shown in Figure 9b, which shows the precipitation pattern. There is a broad

area where precipitation measured between 30 and 40 mm and the maximum values were around 60 mm.

**5. Summary and conclusions**

We have analysed episodes that featured intense orographic precipitation and also had high values in the instability indices that affected the Central range of the Iberian Peninsula during the years 1958 to 2010. The synoptic characteristics associated with the cases analysed were a strong south-westerly tropospheric flow, with jet streams at low

and high levels, and high moisture contents at low levels. This flow was the result of low pressure systems over the North Atlantic area, with associated fronts affecting the area studied. In general, the CAPE values and the Total Totals (TT) and K instability indices indicate static instability, in agreement with the lightning flashes observed, especially windward.



The moisture flux associated with the cases of heavy orographic precipitation considered here was > 100 m g (s kg)$^{-1}$, and both the dry and moist Froude numbers were >1. These numbers can be considered to be characteristic for heavy precipitation in the Central range when they are associated with long-lasting intervals. The broadest and highest part of this mountain range (Gredos range) generates an "orographic shadow" leeward. Precipitation from frontal systems is increased sharply by orographic vertical movement in the Central range, where updrafts associated with the warm conveyor belt are increased. The precipitation pattern in these cases resembles a "C", with heavy precipitation south of the mountain range. In several cases, the total amount of precipitation was moderate. This seems to have been caused by the short time interval where airflow perpendicular to the orographic barrier persisted.

**Author Contribution:** M. Mora and F. de Pablo designed the ideas and concepts of the work and they prepared the manuscript. J. Riesco and L. Rivas carried out the analysis of the data and evaluated the results. J.M. Sánchez developed the entire part graphic and checked the language. All the authors conducted a final review of the manuscript.

**Competing interests:** The authors declare that they have no conflict of interest.

**Acknowledgements:** The authors are grateful to AEMET (Agencia Estatal de Meteorología) for providing the data and partial support for this research. Special thanks to Carlos Jiménez Alonso.

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





**Table captions**

Table 1. Date and values at the 850 hPa pressure level of several meteorological indices for each case considered. Vq = meridional moisture flux [m g (s kg)$^{-1}$]; P = number of 6-hour intervals with Vq higher than the indicated value (70 or 100); F = maximum dry Froude number; Fw = maximum moist Froude number; CAPE, TT and K = maximum CAPE [J/kg], Total Totals, and K instability indices.

Table 2. Maximum (Pmax) and mean (Pmean) precipitation observed in the Gredos range (SGRED) during each case considered. [The Pmean has been calculated using a geographic information system (GIS)]

Table 3. Time variation for several meteorological indices along 27 February 2010. TT and K are the Total Totals and K instability indices; Vq[m g (s kg)$^{-1}$], F, and Fw are the meridional moisture flux, Froude number, and moist Froude number at the 850 hPa pressure level.



TABLE 1.

| Case | Date | Vq Máximum [m g (s kg)$^{-1}$] | Vq Mean | P> 70 | P> 100 | F | Fw | CAPE (Jkg$^{-1}$) | TT (ºC) | K (ºC) |
|---|---|---|---|---|---|---|---|---|---|---|
| 1 | 14-15/11/1963 | 162 | 120 | 6 | 3 | 1.4 | 1.5 | 801 | 52 | 35 |
| 2 | 14-15/01/1975 | 114 | 100 | 8 | 5 | 1.4 | 1.5 | 287 | 51 | 30 |
| 3 | 10/01/1970 | 102 | 102 | 1 | 1 | 1.1 | 1.1 | 619 | 52 | 29 |
| 4 | 26-31/12/1981 | 171 | 103 | 8 | 3 | 1.5 | 1.6 | 534 | 53 | 34 |
| 5 | 19-20/10/2004 | 195 | 135 | 8 | 6 | 1 | 1.1 | 1141 | 47 | 35 |
| 6 | 6-7/11/1982 | 154 | 102 | 7 | 3 | 1.3 | 1.3 | 506 | 54 | 32 |
| 7 | 7-8/01/1992 | 111 | 95 | 4 | 2 | 1 | 1 | 0 | 50 | 30 |
| 8 | 16-18/12/1997 | 168 | 120 | 6 | 4 | 1.1 | 1.1 | 892 | 53 | 37 |
| 9 | 23-25/09/1965 | 203 | 134 | 4 | 3 | 1.3 | 1.4 | 788 | 50 | 36 |
| 10 | 22-25/10/2006 | 206 | 127 | 15 | 11 | 1.3 | 1.4 | 1278 | 47 | 31 |
| 11 | 8-16/10/1993 | 153 | 111 | 18 | 14 | 1.5 | 1.6 | 1517 | 52 | 28 |
| 12 | 17-19/12/1958 | 160 | 124 | 5 | 4 | 1.1 | 1.2 | 816 | 50 | 35 |
| 13 | 2-6/02/1972 | 129 | 87 | 12 | 5 | 1.4 | 1.4 | 600 | 52 | 31 |
| 14 | 14-22/11/1989 | 147 | 112 | 12 | 7 | 1.4 | 1.5 | 640 | 50 | 31 |
| 15 | 2-3/12/1987 | 116 | 101 | 4 | 3 | 1.1 | 1.1 | 140 | 52 | 31 |
| 16 | 23-25/11/2006 | 193 | 144 | 5 | 5 | 1.8 | 1.9 | 261 | 44 | 32 |
| 17 | 4-8/12/2010 | 110 | 91 | 6 | 2 | 0.8 | 0.8 | 821 | 54 | 31 |
| 18 | 21-22/12/2010 | 87 | 85 | 2 | 0 | 0.7 | 0.8 | 497 | 51 | 32 |
| 19 | 27-28/02/2010 | 114 | 103 | 3 | 1 | 1.1 | 1.2 | 441 | 45 | 28 |
| Average | | 147 | 110 | 7 | 4 | 1.2 | 1.3 | 662 | 50 | 32 |





TABLE 2.

| Case | Pmax SGRED (mm) | Pmean SGRED (mm) |
|---|---|---|
| 1 | 310 | 88 |
| 2 | 274 | 96 |
| 3 | 305 | 77 |
| 4 | 504 | 240 |
| 5 | 235 | 98 |
| 6 | 322 | 116 |
| 7 | 291 | 58 |
| 8 | 377 | 145 |
| 9 | 214 | 74 |
| 10 | 317 | 103 |
| 11 | 867 | 271 |
| 12 | 223 | 111 |
| 13 | 406 | 188 |
| 14 | 694 | 300 |
| 15 | 242 | 95 |
| 16 | 310 | 88 |
| 17 | 364 | 136 |
| 18 | 270 | 74 |
| Average | 362 | 131 |





TABLE 3.

| Day/UTC hour | Vq [m g (s kg)$^{-1}$] | TT (ºC) | K (ºC) | F | Fw |
|---|---|---|---|---|---|
| 26/18 | 25 | 38 | 20 | 0.32 | 0.33 |
| 27/00 | 42 | 40 | 19 | 0.33 | 0.33 |
| 27/06 | 119 | 39 | 26 | 0.74 | 0.76 |
| 27/12 | 146 | 45 | 28 | 0.87 | 0.91 |
| 27/18 | 77 | 42 | 20 | 1.09 | 1.2 |
| 28/00 | 12 | 32 | 6 | 0.26 | 0.27 |



**Figure captions**

Figure 1.(a) Main orographic features in the north-western Iberian Peninsula. Polygons represent the provinces of the region of Castilla-Leon. The dashed line corresponds to the area studied. (b) Study area: polygon indicate the Gredos range (SGRED); vertical lines AB is the reference line for the orographic and precipitation profile show in Figure 2; black dots

indicate the weather stations where precipitation was measured; the circle and triangle are respectively the location of the remote-controlled weather stations at La Covatilla and Navarredonda; the star is the reference point for calculating the parameters used in the study (40 ºN, 5 ºW).

Figure 2. Average fields at 12 UTC of (a) mean sea level pressure (hPa); (b) wind (m/s) and (c) specific humidity (g/kg) at the 850 hPa pressure level; (c) geopotential (m) at the 500 hPa pressure level.

Figure 3. Average precipitation (line) and orographic (shaded) profiles along line AB (see Figure 1b).

Figure 4. 23-25 November 2006: (a) infrared channel image from MSG of 24 November at 00UTC; (b) surface pressure pattern of 24 November at 00UTC; (c) 300 hPa wind pattern (lines are isotachs, units kt) of 25 November at 12UTC; and (d) 850 hPa wind pattern (kt) over the Iberian Peninsula of 25 November at 00UTC.

Figure 5. (a) Cloud-to-ground lightning flashes detected (+ positive flashes) on 24 November over the Iberian Peninsula,

and (b) Madrid sounding (40 ºN, 3 ºW) of 24 November at 12UTC.

Figure 6. (a) Temporal distribution of precipitation observed at the La Covatilla weather station (see Figure 1(b)) on 24 November; (b) Spatial pattern of average precipitation observed over the area for the period 1971-2000 including the precipitation of the north of Extremadura.

Figure 7. Spatial (a) and precipitation (b) profiles along a south-north-oriented line crossing the Gredos range.

Figure 8. 27 February 2010: (a) Surface pressure pattern at 12UTC; and (b) Doppler radar image over the Spanish region of Castilla- León. Negative values indicate approaching movement towards the radar position (*). Positive values indicate points are moving away from the radar position.

Figure 9. (a) Temporal distribution of precipitation, wind direction, and wind intensity (maximum and mean) on 27 February observed at the weather station at Navarredonda (see Figure 1b); and (b) spatial pattern of precipitation (isolines

in mm) observed on 27 February over the study area (circle, Navarredonda remote-controlled weather station).





Figure 1.    Mora García et al.,

a)

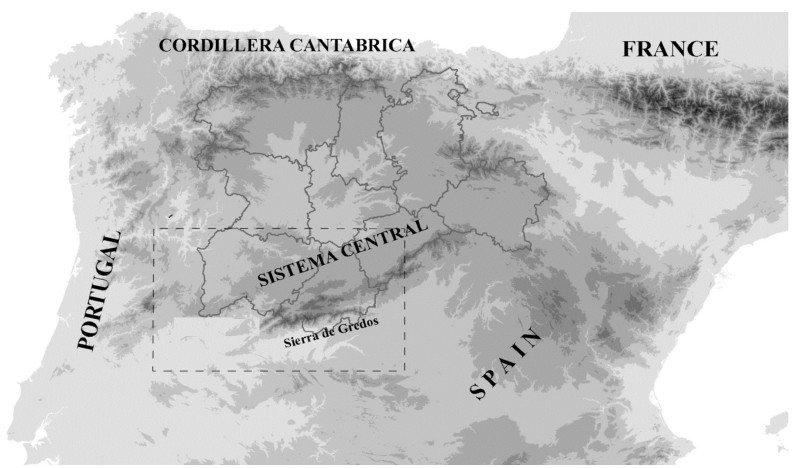

b)

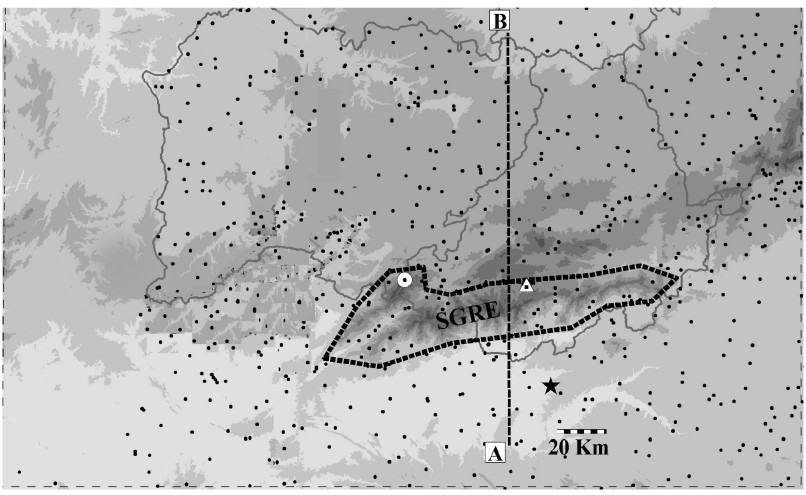



Figure 2. Mora García et al.,

(a)

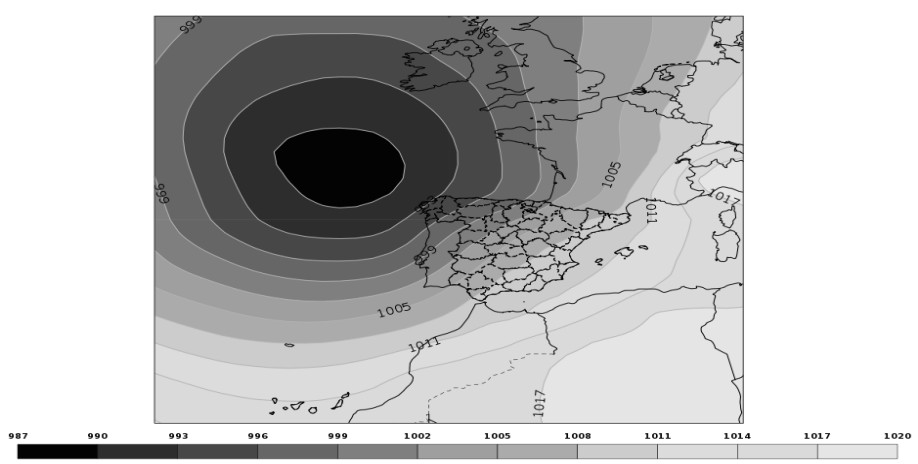

(b)

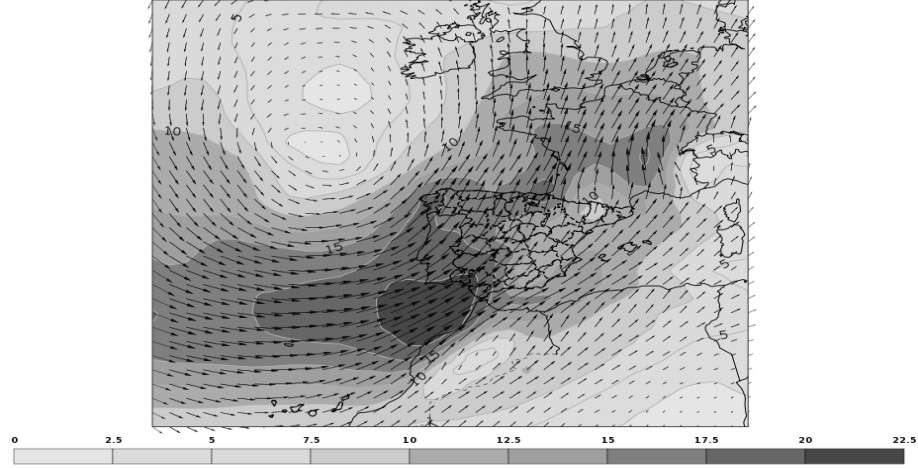




Figure 2. (cont.)          Mora García et al.,

(c)

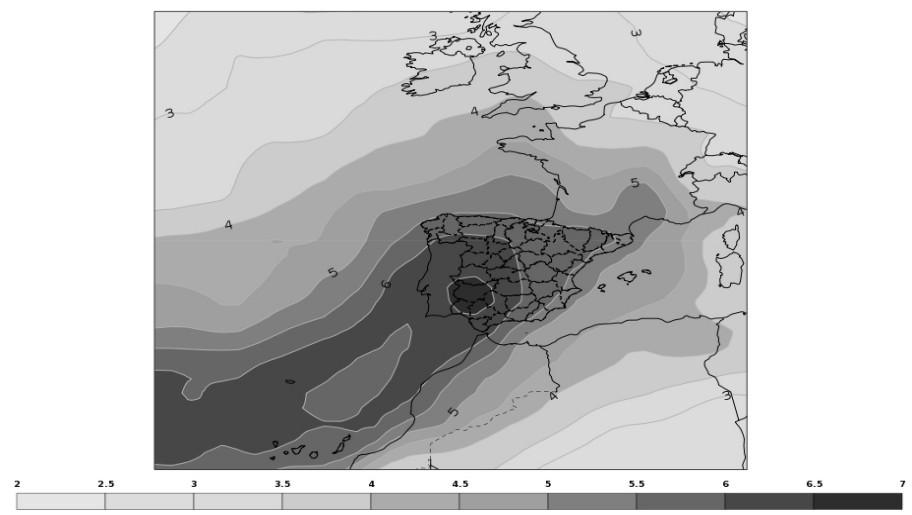

(d)

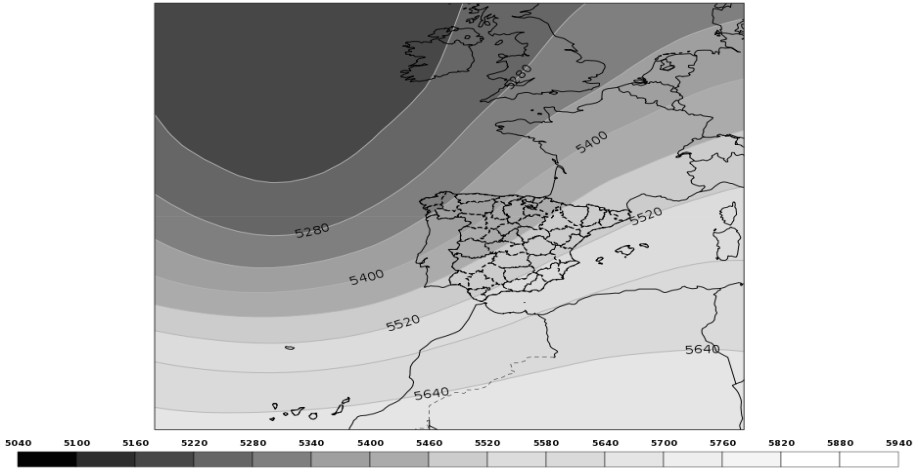



Figure 3.                 Mora García et al.,

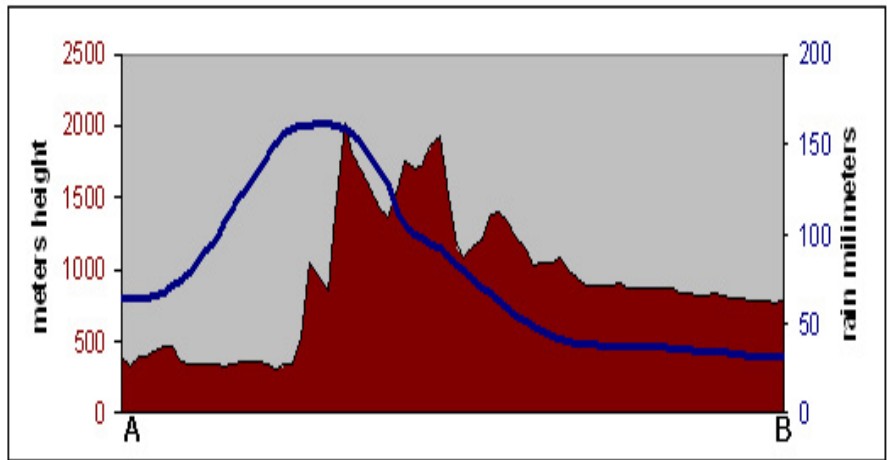


Figure 4.                    Mora García et al.,

a)

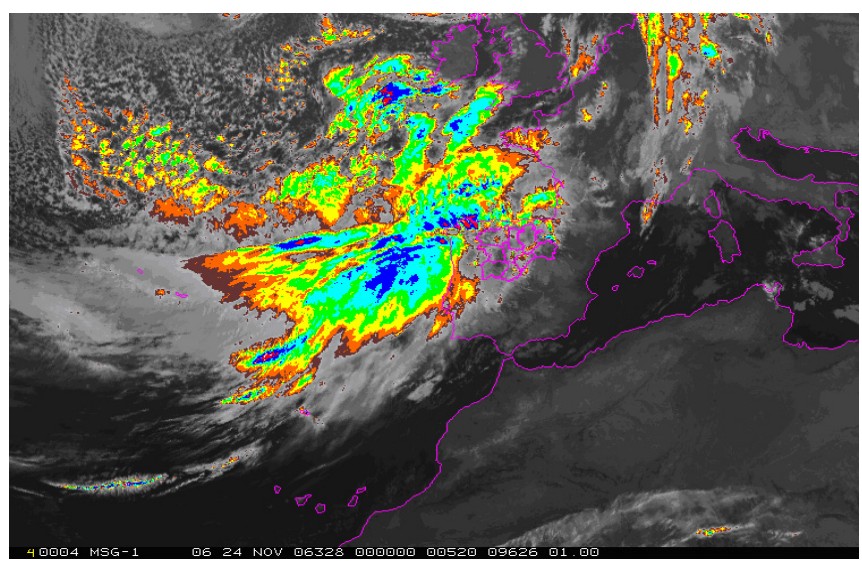

b)

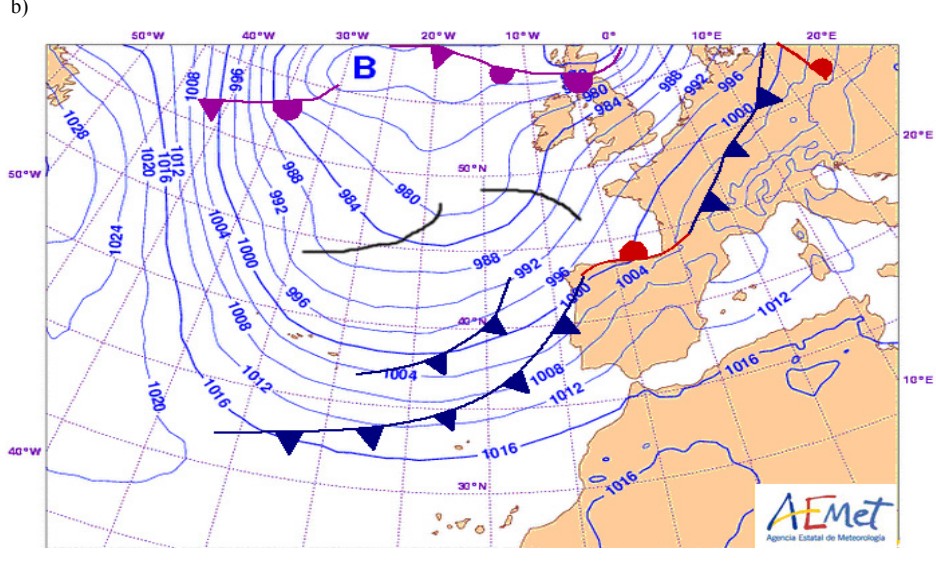





Figure 4 (cont.)        Mora García et al.,

c)

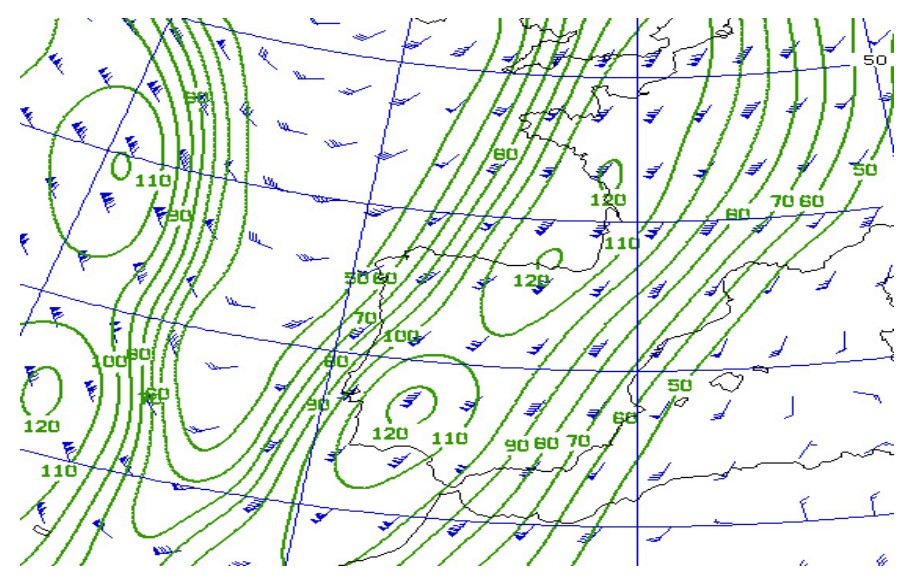

d)

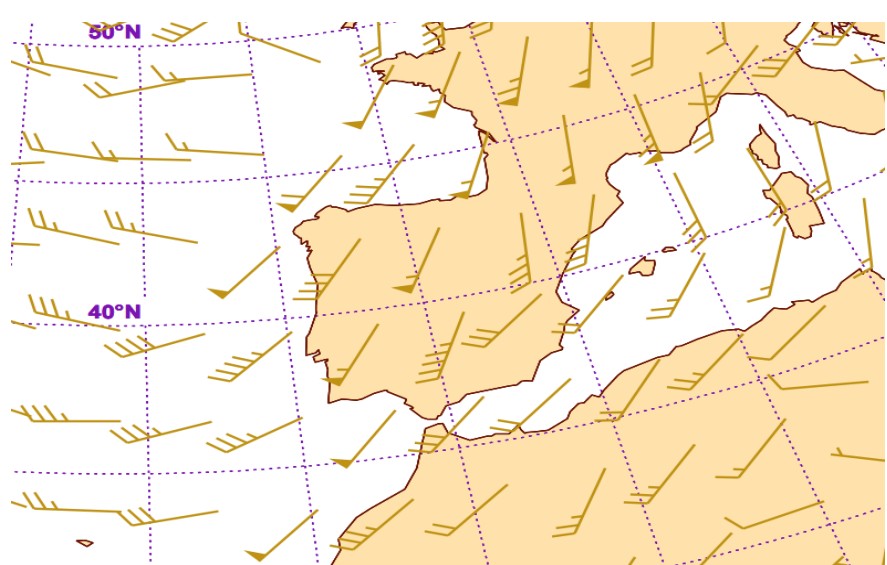



Figure 5.  Mora García  et al.,

a)

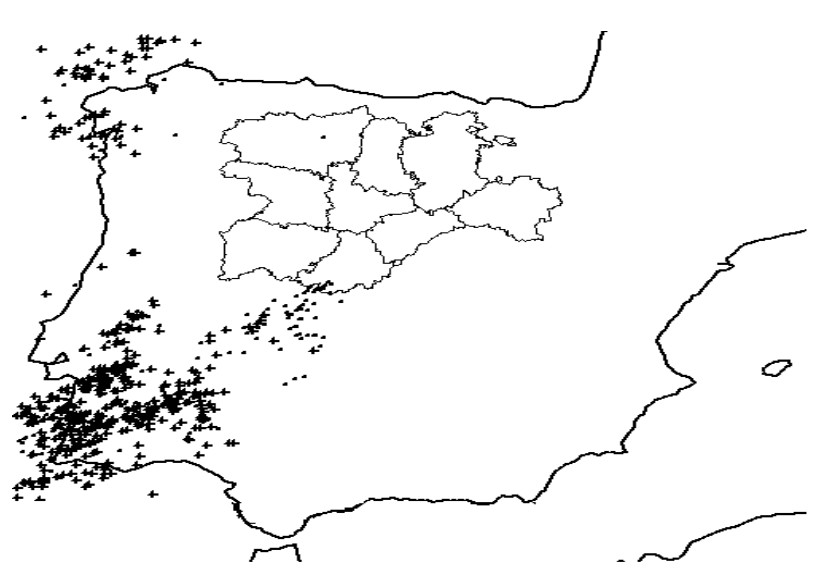

b)

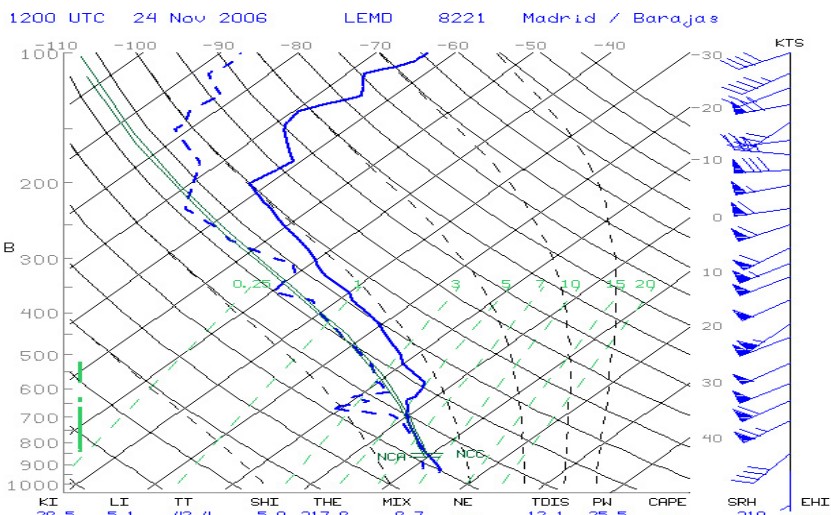




Figure 6.            Mora García et al.,

a)

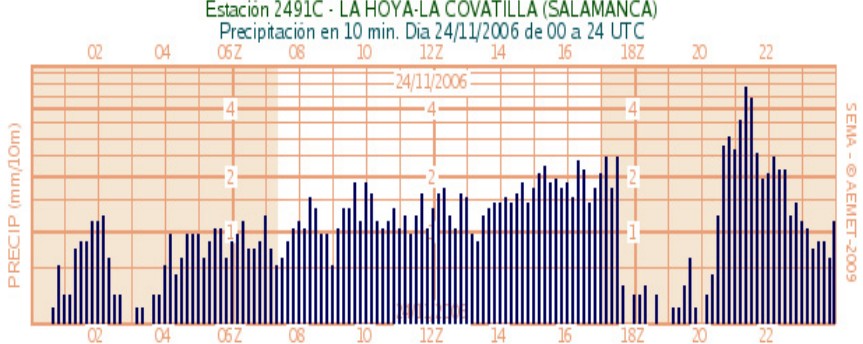

b)

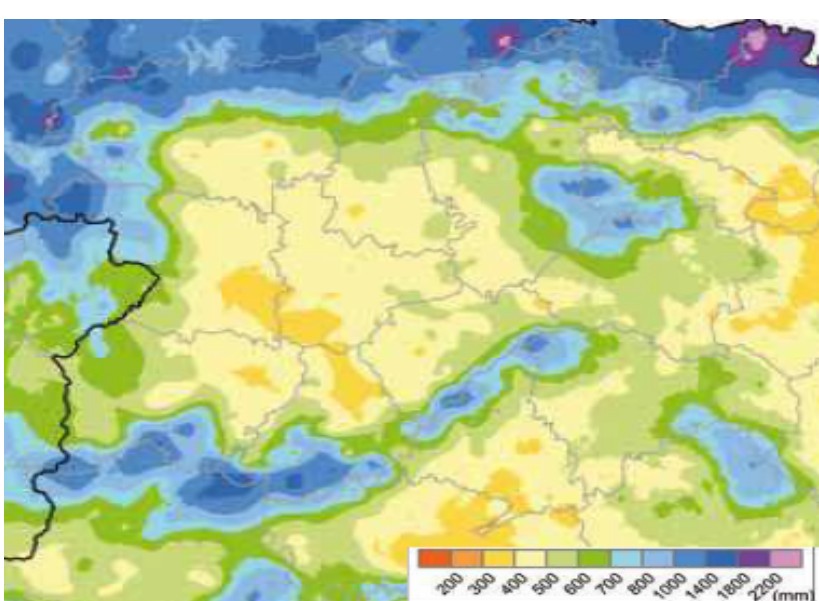



Figure 7.    Mora García et al.,

a)

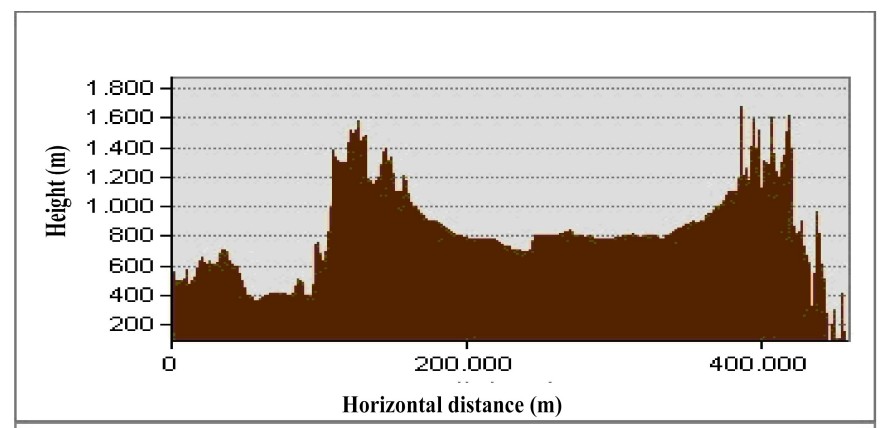

b)

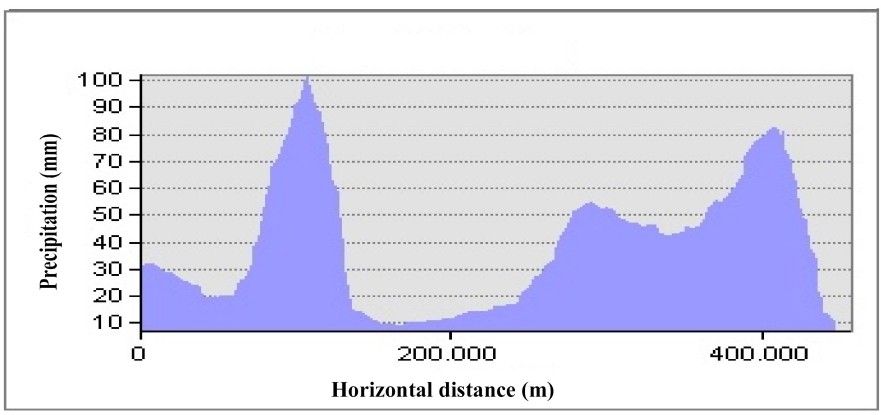




Figure 8.            Mora García et al.,

a)

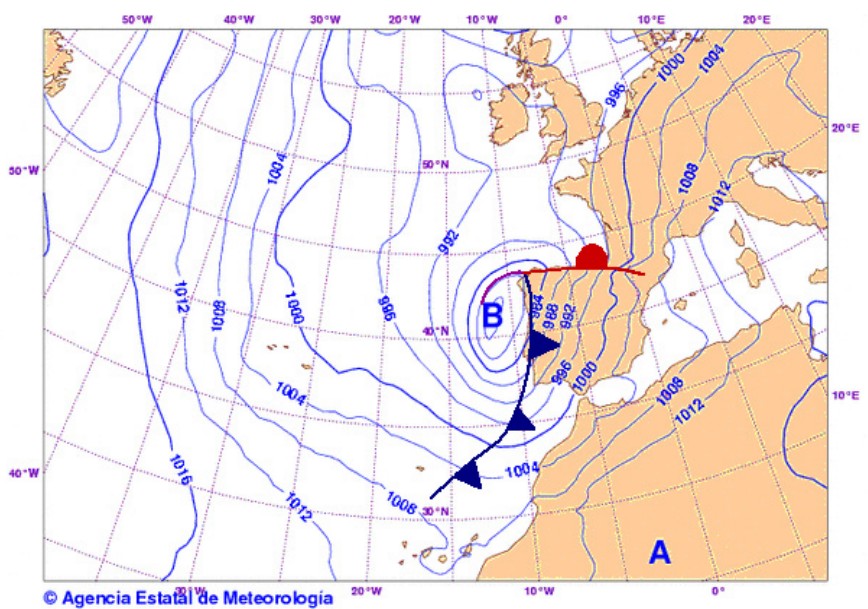

b)

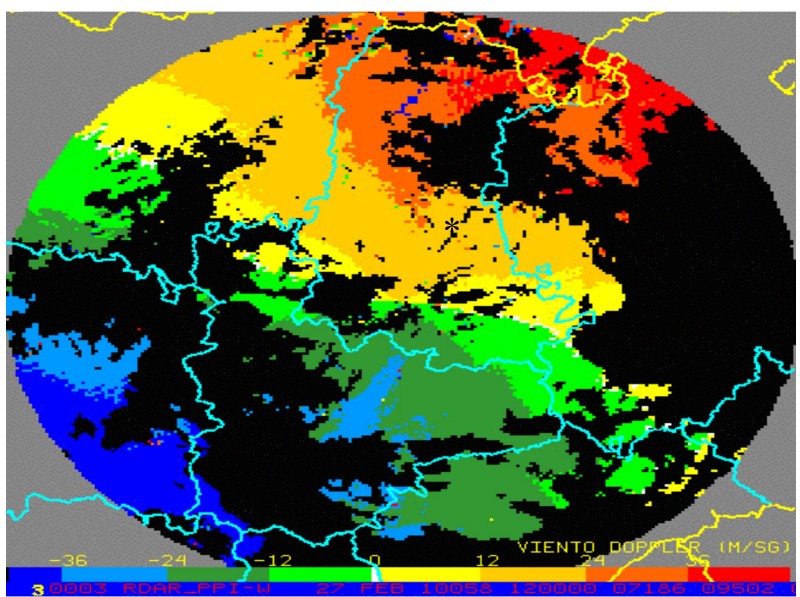





Figure 9.          Mora García et al.,

a)

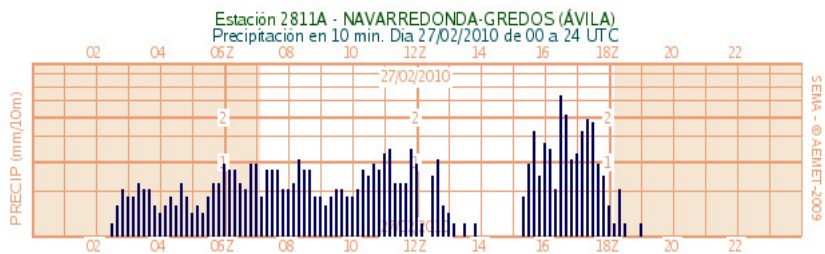

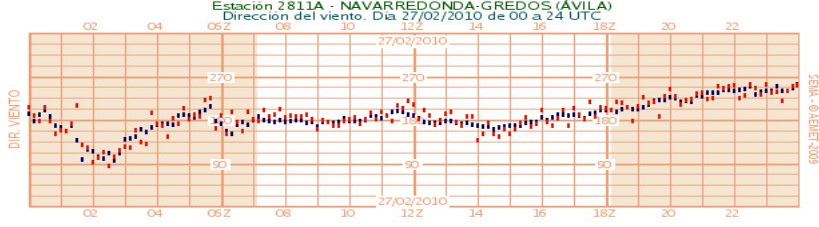

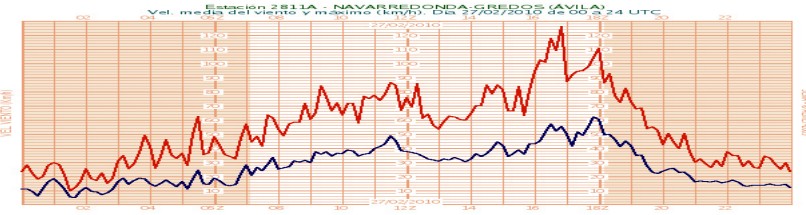

b)

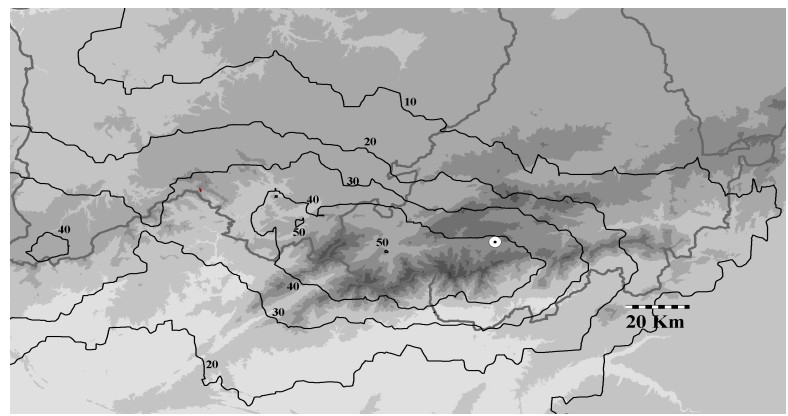