# Peer review of "Intense precipitation events in the Central range of the Iberian Peninsula"

_Natural Hazards and Earth System Sciences, 2017_

## Referee Comment (RC1) · Anonymous Referee #1 · 8 Mar 2017

This article reports about 19 heavy precipitation events at the central Spanish mountain range. Apart from precipitation flow direction, low level wind speed, moisture fluxes and Froude numbers are the main features investigated.

The results of this study are important for regional weather forecasters but are otherwise not new. In their abstract the authors write: "The effect of the Central range on the spatial distribution of precipitation on the Iberian Peninsula plateau results in a sharp increase in precipitation in the 15 south of the Central mountain range, followed by a decrease to the north of this range" - this is really not a new result. While some aspects of their analysis goes beyond classical synoptic reasoning, such as looking at Froude numbers and moisture fluxes, this kind of analysis is not new either. Methodically the analysis is correct, but the large scale data used are not optimal. For the period 1958-1978 the JRA55 reanalysis should be used, which are available since at

least 2014, after that ERA-Interim data are available since 2011 at least. These data sets are much more homogeneous than ERA-40+operational ECMWF data. This is important since moisture fluxes from events in 1961 are difficult to compare with those of more recent events if the analysis methods are so different as they are between ERA-40 and operational ECMWF analyses. The intensity assessment for precipitation relies on a few station data. However the study shows precipitation maps and cross sections (Figs 3,6,7,9b) which must have been produced by some gridding technique. Did the authors just use Kriging or similar as it is available in the ARCGIS Software? There are better tecniques for that (see e.g. Frei and Schär 1998, Isotta 2013).

I am also not happy with the quality of the figures. Many of them (Figs 1,2,4c,4d,5a,6b,8b,9b) seem to be just screen shots cut out from some display, since they do not have proper lat/lon axis frames. That is really below international standards. In Fig. 3 there is no x axis scaling.

A complete revision of the study with improved reanalyis data, improved precipitation analysis and higher figure quality is necessary to lift this manuscript up to international standards. Since this will likely take more time than acceptable for a review process, I recommend rejection of the manuscript, however encouraging resubmission.

---

## Referee Comment (RC2) · Anonymous Referee #2 · 28 Jul 2017

The paper entitled "Intense precipitation events in the Central range of the Iberian Peninsula" assesses various atmospheric stability and moisture indices during heavy precipitation events for the above mentioned mountain range. The study of heavy precipitation indices over mountainous areas is relevant and useful subject. However, at this stage, the manuscript requires significant work before it should be considered for publication.

Main comments:

1. There are issues with regards to the methodology.

a. More details need to be provided on the kriging method, as it will have an impact on the results. Please elaborate which technique and why it was chosen.

b. More details are needed on how the precipitation episodes were defined. From Table 1 the length of the events varies significantly (from less than one day to eight days), which makes it very hard to compare the precipitation totals in Table 2. In the abstract you mention 24 hours, but I cannot find anywhere that this is mentioned again in the text. On page 4, it is mentioned that the episodes were chosen with 'the heaviest precipitation', but over what time period, over what area/which rain gauges? For Case study b, you acknowledge that the total precipitation was less due to the short duration of flow perpendicular to the mountain range, yet, is this 30-40mm in 18 hours similar to other events, less than other events? Calculating the maximum hourly precipitation, or daily precipitation amount could be a solution.

c. The choice of models. By referring to the ECWMF model, implies the forecast product – but there is no reference to confirm, and at which time steps the variables were selected. And are the two datasets really comparable? It would be preferable to either use one dataset over the entire period, or use ERA40 + ERA Interim where there is an overlap and you can assess the differences between the two datasets (1979 – 2002).

d. Not clear why the one particular point for the analysis was chosen. Because the windward slope is steepest? Would the values in Table 1 change much if point what somewhere else?

2. Suggest some reorganizing of the theoretical concepts, study area and data. Only some of the indices are introduced (e.g.., Froude number, although somewhat confusingly the equation is left out until the study area section; the index by Lin et al 2001 is introduced, but not mentioned in the list of indices in the next section, but then is included in the results discussion). It is interesting to compare these indices, so I suggest all the indices that you use are included in the theoretical concepts (at least a brief explanation about what is moderate/high values or a reference to where one can find out), including why they were selected as not all indices are included, such as deep layer shear. To provide an example, you do not consider any wind only measurement

for each episode, yet you discuss wind intensity as being an indicator for convective vs stratiform precipitation (page 6). Furthermore, there is inconsistency on the flow regime types: On page 3, type I-III are all for convective systems (with differences in propagation), yet on page 4 you refer to type II as stratiform.

3. There are inconsistencies between the conclusion and the results. For example on page 8 lines 1 - 2 state that ...."the moisture flux associate with the cases of heavy orographic precipitation considered here was...., and both the dry and moist Froude numbers were >1". Although clearly from Table 1, some of the Froude numbers are less than one. In the abstract, you state "all events were associated with a south-westerly flow, a low level jet..." yet you only consider the composite of 19 events and not the individual events. Did you assess each of the events individually? Even if plots of the individual events are not shown, it would be useful to know that each event was indeed associated with the above synoptic situation.

4. The discussion on page 6 is confusing. Where do the values 1.6km and 35km come from and why are they acceptable? 5km is the resolution of what? The DEM you used in ArcGIS? Similarly, doesn't the type III imply the presence of a convective system propagating similar to the flow, not that it is just a convective system?

5. The language also needs to be improved. Some examples:

a. Page1 lines 18-19, what increases? Do you mean "....the forecasting of precipitation is therefore difficult, particularly for forecasts with coarser spatial and temporal resolution"

b. Page 4 lines 4 - 5: "to increase to higher areas" what do you mean?

c. In the abstract: "from 19 episodes, with the highest average values for the study area, of precipitation accumulated within 24 h, occurring between years 1958-2010" Do you mean "from 19 episodes, which have the highest average 24 hour precipitation amounts in the study area between 1958 and 2010"

6. The figures and tables also require some work (see comment from the previous reviewer). Some additional/specific comments:

a. Table 1: Caption needs improvement. It states "the values at 850hPa", but CAPE is measured only at the surface. Total totals also uses information at other pressure levels. The caption should also indicate that these are the values at the point indicated in Figure 1.

b. Table 2: where is case 19? And what is the time period (for the entire days in Table 1)? It would be useful to know the number of hours considered for each episode.

c. Figure 3 – how was the precipitation calculated (blue line)? Is this average precipitation for the episodes? No horizontal distance on the x-axis (difficult to estimate using figure 1b). Can you use something similar as in 7b?

d. Figure 6 – difficult to determine the location, perhaps include lat/lons. It would be useful to include the horizontal line that use in 7b as well for reference.

e. Figure 7 Where is this profile for? (see comment above). And what is meant by 'precipitation profile'? Average for the episodes, or is it just one episode?

---

## Short Comment (SC1) · 18 Sep 2017

Response to RC1

"...The effect of the Central range on the spatial distribution of precipitation on the Iberian Peninsula plateau results in a sharp increase in precipitation in the 15 south of the Central mountain range, followed by a decrease to the north of this range" - this is really not a new result. ..."

So far, we are not aware that there are similar studies to our for the area of interest.

"...For the period 1958-1978 the JRA55 reanalysis should be used, which are available since at least 2014, after that ERA-Interim data are available since 2011 at least. These data sets are much more homogeneous than ERA-40+ operational ECMWF data..."

We think that the JRA55 reanalysis had an objective to improve information, mainly in the area of Asia. In addition, we do not believe that there is a clear pronouncement of the scientific community on which of the models of reanalysis is the most adequate, or if the JRA55 is better than ERA-INTERIM.

There are numerous studies comparing reanalysis models, but all agree on the great similarity of results. In any case, they present some differences in certain parameters, mainly those that depend strongly on the altitude, seasonality or area of study.

For example, in this reference https://climatedataguide.ucar.edu/climatedata/jra-55 there are some weaknesses presented by JRA55:

Key limitations:

As with most reanalyses, diagnostic variables including precipitation and evaporation should be used with extreme caution.

Dry bias in upper and middle troposphere and in regions of deep convection. Time-varying warm bias in the upper troposphere.

Accordingly, the calculation of the moisture flows is also not very reliable in the JRA55. To corroborate that there is no unanimity in the model to be chosen, we indicate an analysis of the data from Ireland, in which it is not clear which of the models, ERA- Interim, ERA-40 or NCEP, is the most appropriate. http://eprintsprod.nuim.ie/2513/1/MooneyMulliganFealy2011.pdf

"...However the study shows precipitation maps and cross sections (Figs 3,6,7,9b) which must have been produced by some gridding technique. Did the authors just use Kriging or similar as it is available in the ARCGIS Software?

Regarding the grid technique, the one supplied in the Arcmap package (ArcGis) was used. Taking into account the complex orography of the study area, the maps rep-resenting precipitation should be understood as an estimate of the

possible real pre- cipitation field, and we are not aware of any interpolation technique in highly irregular mountainous areas that optimize the representation of the precipitation field.

"...I am also not happy with the quality of the figures. Many of them (Figs  $% \mathcal{F}_{\mathrm{reg}}$

1,2,4c,4d,5a,6b,8b,9b) seem to be just screen shots cut out from some display, since they do not have proper lat/lon axis frames. That is really below international standards. In Fig. 3 there is no x axis scaling. ..."

We will try to improve the quality of the figures.

\*\* After making the corrections proposed by all reviewers, we believe have improved the quality of the figures and have clarified some issues.

---

## Author Comment (AC2) · 18 Sep 2017

The comment was uploaded in the form of a supplement:
https://www.nat-hazards-earth-syst-sci-discuss.net/nhess-2017-76/nhess-2017-76-AC2-supplement.pdf

---

## Author Comment (AC3) · 18 Sep 2017

The comment was uploaded in the form of a supplement:
https://www.nat-hazards-earth-syst-sci-discuss.net/nhess-2017-76/nhess-2017-76-AC3-supplement.pdf

---

## Author Comment (AC4) · 18 Sep 2017

The comment was uploaded in the form of a supplement:
https://www.nat-hazards-earth-syst-sci-discuss.net/nhess-2017-76/nhess-2017-76-AC4-supplement.pdf

---

## Author Response (AR2)

Response to RC1

"...The effect of the Central range on the spatial distribution of precipitation on the Iberian Peninsula plateau results in a sharp increase in precipitation in the 15 south of the Central mountain range, followed by a decrease to the north of this range" - this is really not a new result. ..."

So far, we are not aware that there are similar studies to our for the area of interest.

"...For the period 1958-1978 the JRA55 reanalysis should be used, which are available since at least 2014, after that ERA-Interim data are available since 2011 at least. These data sets are much more homogeneous than ERA-40+ operational ECMWF data. ..."

We think that the JRA55 reanalysis had an objective to improve information, mainly in the area of Asia. In addition, we do not believe that there is a clear pronouncement of the scientific community on which of the models of reanalysis is the most adequate, or if the JRA55 is better than ERA-INTERIM.

There are numerous studies comparing reanalysis models, but all agree on the great similarity of results. In any case, they present some differences in certain parameters, mainly those that depend strongly on the altitude, seasonality or area of study.

For example, in this reference https://climatedataguide.ucar.edu/climate-data/jra-55 there are some weaknesses presented by JRA55:

Key limitations:
  As with most reanalyses, diagnostic variables including precipitation and evaporation should be used with extreme caution.
  Dry bias in upper and middle troposphere and in regions of deep convection.
  Time-varying warm bias in the upper troposphere.

Accordingly, the calculation of the moisture flows is also not very reliable in the JRA55. To corroborate that there is no unanimity in the model to be chosen, we indicate an analysis of the data from Ireland, in which it is not clear which of the models, ERA- Interim, ERA-40 or NCEP, is the most appropriate. http://eprintsprod.nuim.ie/2513/1/MooneyMulliganFealy2011.pdf

"...However the study shows precipitation maps and cross sections (Figs 3,6,7,9b) which must have been produced by some gridding technique. Did the authors just use Kriging or similar as it is available in the ARCGIS Software?

Regarding the grid technique, the one supplied in the Arcmap package (ArcGis) was used. Taking into account the complex orography of the study area, the maps rep- resenting precipitation should be understood as an estimate of the

possible real pre- cipitation field, and we are not aware of any interpolation technique in highly irregular mountainous areas that optimize the representation of the precipitation field.

"…I am also not happy with the quality of the figures. Many of them (Figs 1,2,4c,4d,5a,6b,8b,9b) seem to be just screen shots cut out from some display, since they do not have proper lat/lon axis frames. That is really below international standards. In Fig. 3 there is no x axis scaling. .."

We will try to improve the quality of the figures.

** After making the corrections proposed by all reviewers, we believe have improved the quality of the figures and have clarified some issues.

Response to RC2

Main comments:

1. There are issues with regards to the methodology.

a. More details need to be provided on the kriging method, as it will have an impact on the results. Please elaborate which technique and why it was chosen.
Precipitation is very difficult to interpolate in mountainous areas, so that the impact of the interpolation method on the area-averaged precipitation is small and, in consequence we used the kriging interpolation method because is the de default method in the ArcGIS© software.
For more information,
[https://desktop.arcgis.com/es/arcmap/latest/extensions/geostatistical-analyst/understanding-ordinary-kriging.htm]

b. More details are needed on how the precipitation episodes were defined. From Table 1 the length of the events varies significantly (from less than one day to eight days), which makes it very hard to compare the precipitation totals in Table 2. In the abstract you mention 24 hours, but I cannot find anywhere that this is mentioned again in the text. On page 4, it is mentioned that the episodes were chosen with 'the heaviest precipitation', but over what time period, over what area/which rain gauges? For Case study b, you acknowledge that the total precipitation was less due to the short duration of flow perpendicular to the mountain range, yet, is this 30-40mm in 18 hours similar to other events, less than other events? Calculating the maximum hourly precipitation, or daily precipitation amount could be a solution.

An episode was selected if the precipitation accumulated in 24 hours had values > 100 mm in at least one observatory inside the selected area. Table 2 shows the values of accumulated precipitation in the observatory that reported the maximum value. Following your recommendation, we calculated the maximum daily (new column in table 2). The case study b is not an episode of intense precipitation due to its short duration. This was precisely what we wanted to show. The data of this episode were taken into account in the averages by mistake. This has been corrected in the revised manuscript.

c. The choice of models. By referring to the ECWMF model, implies the forecast product – but there is no reference to confirm, and at which time steps the variables were selected. And are the two datasets really comparable? It would be preferable to either use one dataset over the entire period, or use ERA40 + ERA Interim where there is an overlap and you can assess the differences between the two datasets (1979 –2002).

Similar Reply to RC1

". . .For the period 1958-1978 the JRA55 reanalysis should be used, which are available since at least 2014, after that ERA-Interim data are available since 2011 at least. These data sets are much more homogeneous than ERA-40+ operational ECMWF data. . ."

We think that the JRA55 reanalysis had an objective to improve information, mainly in the area of Asia. In addition, we do not believe that there is a clear pronouncement of the scientific community on which of the models of reanalysis is the most adequate, or if the JRA55 is better than ERA-INTERIM.

There are numerous studies comparing reanalysis models, but all agree on the great similarity of results. In any case, they present some differences in certain parameters, mainly those that depend strongly on the altitude, seasonality or area of study.

For example, in this reference https://climatedataguide.ucar.edu/climate-data/jra-55 there are some weaknesses presented by JRA55:

Key limitations:
 As with most reanalyses, diagnostic variables including precipitation and evaporation should be used with extreme caution.
 Dry bias in upper and middle troposphere and in regions of deep convection
 Time-varying warm bias in the upper troposphere

Accordingly, the calculation of the moisture flows is also not very reliable in the JRA55. To corroborate that there is no unanimity in the model to be chosen, we indicate an analysis of the data from Ireland, in which it is not clear which of the models, ERA-Interim, ERA-40 or NCEP, is the most appropriate. http://eprintsprod.nuim.ie/2513/1/MooneyMulliganFealy2011.pdf

Moreover, the meteorological fields used in this study (wind and moisture) are in the pressure levels of 850 and 500 hPa, that is, above the PBL. This means that the horizontal resolution has not a great impact.

d. Not clear why the one particular point for the analysis was chosen. Because the windward slope is steepest? Would the values in Table 1 change much if point what somewhere else?

The particular point was chosen because in that point: i) the Gredos range reaches its maximum altitude and, consequently the southerly wind must surpass a big difference in height (1500m), and ii) the slope is maximum. The values in table 1 will be not very different is other points were chosen, because the meteorological fields are outside the

PBL. Data of precipitation will be different in other point, but we wanted to show the values where the orographic effect is expected to be greater.

2. Suggest some reorganizing of the theoretical concepts, study area and data. Only some of the indices are introduced (e.g.., Froude number, although somewhat confusingly the equation is left out until the study area section; the index by Lin et al 2001 is introduced, but not mentioned in the list of indices in the next section, but then is included in the results discussion). It is interesting to compare these indices, so I suggest all the indices that you use are included in the theoretical concepts (at least a brief explanation about what is moderate/high values or a reference to where one can find out), including why they were selected as not all indices are included, such as deep layer shear. To provide an example, you do not consider any wind only measurement for each episode, yet you discuss wind intensity as being an indicator for convective vs stratiform precipitation (page 6). Furthermore, there is inconsistency on the flow regime types: On page 3, Type I-III are all for convective systems (with differences in propagation), yet on page 4 you refer to type II as stratiform.

We agree and we have reorganized the manuscript.

3. There are inconsistencies between the conclusion and the results. For example on page 8 lines 1– 2 state that . . .”the moisture flux associate with the cases of heavy orographic precipitation considered here was. . .., and both the dry and moist Froude numbers were >1”. Although clearly from Table 1, some of the Froude numbers are less than one. In the abstract, you state “all events were associated with a south-westerly flow, a low level jet. . .” yet you only consider the composite of 19 events and not the individual events. Did you assess each of the events individually? Even if plots of the individual events are not shown, it would be useful to know that each event was indeed associated with the above synoptic situation.

Lines 1-2 on page 8 have been changed. The characteristics of each individual event are similar to those of the average (this is indicated in the revised manuscript). It should be noted that the inclusion of synoptic maps for individual events would make the paper excessively large.

4. The discussion on page 6 is confusing. Where do the values 1.6km and 35km come from and why are they acceptable? 5km is the resolution of what? The DEM you used in ArcGIS? Similarly, doesn't the Type III imply the presence of a convective system propagating similar to the flow, not that it is just a convective system?

The mean slope is 0.05 (which is now indicated in section 2 page 4, line 17) ≈ 16/35. The value 0.09 is the maximum slope. This is corrected in the revised manuscript. Line 7, page 6 is also corrected

5. The language also needs to be improved. Some examples:

a. Page1 lines 18-19, what increases? Do you mean ". . ..the forecasting of precipitation is therefore difficult, particularly for forecasts with coarser spatial and temporal resolution"
The text has been corrected.

b. Page 4 lines 4 – 5: "to increase to higher areas" what do you mean?
The text has been reordered and changed

c. In the abstract: "from 19 episodes, with the highest average values for the study area, of precipitation accumulated within 24 h, occurring between years 1958-2010" Do you mean "from 19 episodes, which have the highest average 24 hour precipitation amounts in the study area between 1958 and 2010"
The text has been corrected.

6. The figures and tables also require some work (see comment from the previous reviewer). Some additional/specific comments:

In accordance with suggestions from the Reviewers, some figures have been improved.

a. Table 1: Caption needs improvement. It states "the values at 850hPa", but CAPE is measured only at the surface. TT also uses information at other pressure levels. The caption should also indicate that these are the values at the point indicated in Figure 1.
The caption of Table 1 has been corrected.

b. Table 2: where is case 19? And what is the time period (for the entire days in Table1)? It would be useful to know the number of hours considered for each episode.
Case 19 has been added to Table 1 and Table 2, but is not taken into account to calculate the average value, since it was a singular case with little precipitation.
Unfortunately it is impossible to know the number of hours considered for each episode and we can only assign a minimum period of 24 hours, as shown in Table 2.

c. Figure 3 – how was the precipitation calculated (blue line)? Is this average precipitation for the episodes? No horizontal distance on the x-axis (difficult to estimate using figure 1b). Can you use something similar as in 7b?
Figure 3 represents a cross-section of average precipitation of all events (line) and an orographic profile along the line AB, described in Figure 1b.

d. Figure 6 – difficult to determine the location, perhaps include lat/lons. It would be useful to include the horizontal line that use in 7b as well for reference.
Unfortunately the quality of this figure can not be improved. According to the comment we have added the latitude and longitude to the figure, for a better understanding.

e. Figure 7 Where is this profile for? (see comment above). And what is meant by 'precipitation profile'? Average for the episodes, or is it just one episode?
Figure 7 refers to the case study from 23 to 25 November 2006, and it shows a cross-section of the altitude and precipitation vs horizontal distance.

Response to RC3

General Comments

For example I suggest that in the introduction authors could put into context the region studied with recent article by Alvarez-Rodriguez et al (2007) - see references below- and the results, at least the case studies described, could be compared with the absolute maximum precipitation fit lines for different time periods given by Gonzalez and Bech (2017), either for Spain or for specific Spanish provinces.
Cited references have been incorporated into the text.

Specific Comments

1. Page 2, line 3. (Now, line 7 on page 2) Prat and Barros (2010): reference not found in references section. Please check and add it.
   The reference has been added.

2. Page 4, line 12. Suggest: comparable to convective -> comparable to those from convective origin
   The text has been corrected.

3. Page 4, lines 14 & 15 (and elsewhere in the text). Check English: regimen -> regime
   The text has been corrected.

4. Page 4, line 21 (and elsewhere). Please check units of the amount given (4.7).
   Values (units) have been fixed.

5. Page 4, line 25. m.a.s.l. -> m a.s.l.
   The text has been corrected.

6. Page 4, line 31. Please clarify the selection method of the events. Is it 100 mm in 24h or during which period?
   The selection of the events has clarified in the text.

7. Page 5, line 6. Middle -> Medium
   The text has been corrected.

8. Page 5, line 7. interval. -> interval [remove "." before the URL in brackets]
   The text has been corrected.

9. Page 6, line 8. For consistency, please use Type in capital letters if you refer to a specific type (Type I, Type II, etc.) as in line 6.
   The text has been corrected.

10. Page 6, line 13. Suggest: static stability low and the mountain barrier narrow-> static stability is low and the mountain barrier is narrow
    The text has been corrected.

11. Page 6, line 14. (Now, lines 28-30 on page 5) This sentence is a bit confusing. What about: of the flow in the
    mountains -> of the flow perpendicular to the mountains ?
    Usually considered the intensity of wind perpendicular to the mountains, therefore is not indicated the direction, only intensity. These episodes are always given with a SW wind.

12. Page 6, line 14. I suggest: cause -> favour, because in fact it depends on the stability conditions
The text has been corrected.

13. Page 7, line 1. (Now, line 17 on page 6) I think additional decimal digits should be given for the Madrid sounding location.
The text has been corrected.

14. Page 7, line 2 (and elsewhere in the text). Suggest: remote-controlled station ->automatic [I do not think that being remote-controlled is relevant]
The text has been corrected.

15. Page 7, line 17. Hickey, 2011: reference not listed in references section.
The reference has been added.

16. Page 11, Table 3. caption indices along 27 -> indices along 26, 27 and 28
The text has been corrected.

17. Page 11, Table 3 caption. Clarify in the caption which variables listed refer to 850hPa level.
The caption of Table 3 has been corrected.

18. Page 12, Table 1. Typo: Máximum -> Maximum (without accent)
The text has been corrected.

19. Page 12, Table 1. Units should be given also for Vq mean.
The text has been corrected.

20. Page 13, Table 2. Suggest adding more columns with the maximum precipitation in 24h and other periods such as 1h, 3h, 6h or 12h; I strongly recommend at least including the 24h; the 1h value may be useful to assess the convective character of the event. Values currently listed are difficult to compare as may correspond to different time periods.
Added a new column to the Table 2, with the values of the maximum precipitation in 24 hours.

21. Page 15, line 3, Figure 1 caption. Show- shown
The text has been corrected.

22. Page 15, Figure 2 caption. Please add: Average fields -> Average fields for the episodes studied (listed in Table 1)
The text has been corrected.

23. Page 15, Figure 3 caption. Average precipitation for which time period? All the event?
It corresponds to the average precipitation of all events. Added to the caption of the Figure 3.

24. Page 15, Figure 4 caption. Suggest: units -> labelled in kt
The text has been corrected.

25. Page 15, Figure 7 caption. Spatial -> Topographic
The text has been corrected.

26. Page 15, Figure 8 caption. Doppler radar image -> Doppler radar wind (m/s) PPI image [you can expand PPI into Plan Position Indicator if preferred]
The text has been corrected.

27. Figure 2. I suggest to improve the panels by removing the current titles above each panel (the labels a,b.. should suffice) and also by redrawing the legend bar to fit the width of each panel. This should allow a more compact and clear display.
According to the comment, the Figure 2 has been rectified.

28. Figure 3. Please improve the quality of the image (resolution, units in brackets).
According to the comment, the Figure 3 has been rectified.

29. Figure 4, Could it be possible to add a colour legend for the cloud top temperatures? Coldest values could be commented in the text.
According to the comment, we have added a colour legend.

30. Figure 6b. Please improve resolution.
Unfortunately the quality of this figure can not be improved. The inclusion of this figure, (average precipitation 1971-2000), serves to confirm that the study area presents a clear orographic influence on rainfall.

31. Figure 7. Regarding the x-axis units labels note that you are using a dot "." which in English usually means decimal separator. Presumably the label 400.000 m means 400 km, does'nt it? Please check and make necessary corrections to avoid confusions.
According to the comment, the Figure 7 has been rectified.

32. Figure 8b. The star symbol seems to be wrongly placed - it is not at the centre of the PPI image - it seems to me it should be further south-west from the current position.

Unfortunately the Doppler radar PPI image is slightly distorted, so that the symbol should be a circle and not a star, which corresponds to the position where the radar is located. The radar is located in Autilla del Pino (41.99°N 4.63°W) about 200 km from the study area, but its image is representative of the average synoptic flow, considering that it is in a particularly flat area. This symbol should not be confused with the point of grid that we have selected to perform reanalysis.